# GA−Reinforced Deep Neural Network for Net Electric Load Forecasting in Microgrids with Renewable Energy Resources for Scheduling Battery Energy Storage Systems

Chaoran Zheng, Mohsen Eskandari *, Ming Li and Zeyue Sun

School of Electrical Engineering and Telecommunications, University of New South Wales, Sydney, NSW 2052, Australia
* Correspondence: m.eskandari@unsw.edu.au

**Abstract:** The large−scale integration of wind power and PV cells into electric grids alleviates the problem of an energy crisis. However, this is also responsible for technical and management problems in the power grid, such as power fluctuation, scheduling difficulties, and reliability reduction. The microgrid concept has been proposed to locally control and manage a cluster of local distributed energy resources (DERs) and loads. If the net load power can be accurately predicted, it is possible to schedule/optimize the operation of battery energy storage systems (BESSs) through economic dispatch to cover intermittent renewables. However, the load curve of the microgrid is highly affected by various external factors, resulting in large fluctuations, which makes the prediction problematic. This paper predicts the net electric load of the microgrid using a deep neural network to realize a reliable power supply as well as reduce the cost of power generation. Considering that the backpropagation (BP) neural network has a good approximation effect as well as a strong adaptation ability, the load prediction model of the BP deep neural network is established. However, there are some defects in the BP neural network, such as the prediction effect, which is not precise enough and easily falls into a locally optimal solution. Hence, a genetic algorithm (GA)−reinforced deep neural network is introduced. By optimizing the weight and threshold of the BP network, the deficiency of the BP neural network algorithm is improved so that the prediction effect is realized and optimized. The results reveal that the error reduction in the mean square error (MSE) of the GA–BP neural network prediction is 2.0221, which is significantly smaller than the 30.3493 of the BP neural network prediction. Additionally, the error reduction is 93.3%. The error reductions of the root mean square error (RMSE) and mean absolute error (MAE) are 74.18% and 51.2%, respectively.

**Keywords:** backpropagation (BP); electric load prediction; genetic algorithm (GA); microgrids; neural network; renewable energy resources (RESs)

## 1. Introduction

### 1.1. Background

With the remarkable improvements in human living standards that have increased electricity demand, various defects of super-large-scale power systems have become increasingly prominent [1,2]. The power generation of traditional fossil fuel-based power plants cannot efficiently meet the increase in electricity demand. The microgrid concept was proposed in the early 21st century for the integration of clean renewable energy resources (RESs) [3,4]. The microgrid is a small-scale local power system that is composed of distributed energy resources (DERs), control systems, and electric loads [5–7]. The power generation of the microgrid, i.e., DERs, includes renewable energy resources (RERs) and energy storage systems (ESSs) [8–10]. The microgrid is regarded as a new type of modern active power distribution system for the utilization and development of renewable energy [11]. Additionally, it has become an indispensable and powerful supplement to the

large power grid. Now, it has gradually become a vital and effective method to solve the disadvantages of the power system in many countries.

The ESSs are the core of the microgrid. These can guarantee power quality and microgrid reliability as well as reduce energy loss. Among all kinds of energy storage technologies, battery energy storage systems (BESSs) have gradually become a very attractive and prominent technology [12]. Its versatility, fast response speed, high energy density, and high efficiency are the main reasons. By absorbing power from the grid during off-peak hours and supplying it during peak hours, a BESS enables peak shifting/shaving, improves power quality, and alleviates congestion. As a result, various types of BESS are becoming increasingly integrated into modern energy systems. However, despite continuous advances in electrochemical technology, the management and control of BESS remain challenging problems [10].

The problem of how effectively maintaining the power generation and consumption balance impacts the stable operation of the microgrid [12]. If the network topology or load of the microgrid changes, appropriate control strategies must be adopted for the power generation device to ensure its safe and reliable operation [13,14]. The output of wind power, photo-voltaic, and other micro-power sources in the microgrid is very unstable, and requires the coordination and cooperation of all DERs [15]. Meanwhile, load forecasting, renewable power generation prediction, and electricity price forecasting are also important factors affecting the performance of the management system in the microgrid [5]. Therefore, it is an important prerequisite for realizing the intelligent management of microgrids [16]. If the load, power, and electricity price of the microgrid can be relatively accurately predicted, the stability and power quality of the power system can be guaranteed.

### 1.2. Current Research on Load Prediction

Many experts and scholars have made a lot of contributions to the development of short-term load forecasting. Generally, the means of prediction could be classified as traditional methods, classical prediction methods, and artificial intelligence prediction methods. Classical methods include the exponential smoothing method and time series method. Traditional methods include the trend extrapolation method and grey prediction method. Intelligent prediction methods include the artificial neural network (ANN) method, chaos theory method, SVM method, and combination prediction method [17]. ANN methods are briefly introduced below.

The principle of ANN is proposed by studying the way of conveying neural signals in the brain. The solving model of the problem is established by a computer, and then the "neurons" are connected by weights according to the simulation of the brain learning process. By constantly changing the weight of the connection through the computer, the established model continuously approximates the nonlinear function of load prediction. Since Park, D.C. et al. [18] first applied an ANN to load prediction in 1991, experts and scholars have continuously deepened their research on neural networks in load prediction and have made remarkable achievements.

For an object with a more complex mapping relationship, ANN can also be used to describe it better and has a strong adaptive ability, so a large number of irregular data can also be processed through its adaptive ability. The disadvantage of the ANN is that it will have a slow convergence speed, and fall into local optimum solutions and overfitting issues.

In [19], the dynamic correction method was proposed to evaluate the weights of neural networks, and the accuracy of load prediction can be effectively improved by using such a correction method. The authors in [20] combined the Hadoop architecture of database technology in the computer field with backpropagation (BP) neural networks. This has also been applied in a study using massive original data to realize load prediction. The combination of the two methods cannot only improve the accuracy of load prediction but also break through the scale of load prediction data. In [21], a short-term load forecasting model based on support vector regression (SVR) and the whale optimization algorithm (WOA) was proposed. However, the initial value of the original WOA algorithm lacks equi-

librium stability. It is possible to fall into local optimum and low convergence accuracy. In literature [22], the fruit fly optimization algorithm (FOA) was improved, and the improved FOA (IFOA) algorithm was combined with the backpropagation neural network (BPNN) to build a wind power generation prediction model. The results show that the signal strength decreases and the packet loss rate increases with the increase in the transceiver distance, and the electromagnetic wave of the wind power plant will cause some interference with the signal strength. The model has limitations on the signal receiving and receiving distance and a high requirement in terms of equipment professionalism.

*1.3. Contribution of the Paper*

This paper aimed to improve the accuracy and effect of the load forecasting module in microgrids. Firstly, after analyzing the operation strategy of BESSs and introducing the structure and importance of the forecasting system, the concept of the BP deep neural network is used to realize the function of load forecasting. Then, considering that the BP neural network easily falls into local optimal solution and convergence speed is slow, a method of improving the BP neural network by genetic algorithm (GA) is proposed. The accuracy of prediction is improved by reducing the input dimension of the BP neural network and optimizing the weight and threshold. Finally, the load forecasting model of the microgrid based on a GA–BP neural network is established to achieve the improvement of prediction accuracy and prediction effect.

## 2. Operational Strategy of BESS and Prediction System

The microgrid can be operated off-grid and grid-connected [23,24]. As shown in Figure 1, the main components of the microgrid are distributed generations (DG), including photovoltaic (PV), wind turbines (WT), fuel cells, (FC), diesel generators (DG), and BESSs [7].

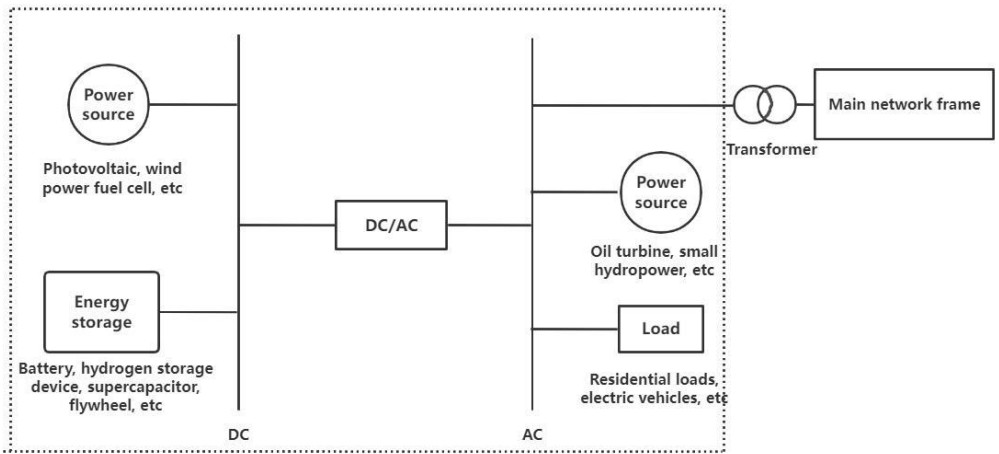

**Figure 1.** Framework of energy management system in microgrid.

*2.1. Operational Strategy of BESS*

The energy management system (EMS) optimizes the operation of the microgrid and schedules the power resources and BESSs in the microgrid, considering fluctuations in renewable energy resources, the uncertainty of market price and demand, and the real−time supply and demand balance of electric energy and other constraints in microgrids [25–27]. Optimization methods are used to reduce operating costs, increase energy efficiency, and reduce the carbon emissions and peak load [28]. In the grid−connected mode, RESs supply loads. As soon as the supply of load power cannot be fulfilled with RESs, it will determine whether to choose battery or the distributed grid to supply according to the state of charge (SOC) of BESS [12]. When the wind and PV power generation have a surplus after load supply, it will also determine whether to sell electricity to the distribution grid or charge the battery according to the SOC of the BESS.

A considered model of the SOC of BESS is shown as follows. The SOC can be described as

$$SoC_t = SoC_{t-1} + \eta * P_{ch,t} - \frac{1}{\eta} * P_{dis,t} \qquad (1)$$

where $\eta$ is the BESS efficiency of charging/discharging; $P_{ch,t}$ is the charging power; and $P_{dis,t}$ is the discharging power. Under ideal conditions, the constraint is

$$0 \leq P_{ch,t} \leq P_b \qquad (2)$$

$$0 \leq P_{dis,t} \leq P_b \qquad (3)$$

where $P_b$ is the installed power of BESS. After considering that the charging and discharging process cannot be carried out simultaneously, there must be another constraint which is

$$P_{ch,t} * P_{dis,t} = 0 \qquad (4)$$

The charge and discharge depth should be considered as

$$\alpha_1 E_b \leq SOC_t \leq \alpha_2 E_b \qquad (5)$$

where $\alpha_1$ and $\alpha_2$ represent the lower and upper limits of BESS; $E_b$ represents the energy capacity of BESS.

The following objective function is considered:

$$OF: \min_{P_{Gp}, P_{DER}, P_{BESS}} \sum_h \left[ (\rho_G P_G - \rho_L P_L) + \sum_{n_{DER}} \left( f_{gen}(P_{DER}) + f_{OM}(P_{DER}) \right) + \sum_{n_{BESS}} \left( f_{BESS}(P_{BESS}) \right) \right] \qquad (6)$$

s.t. (1)–(5);

$$P_G + \sum_{n_{DER}} (P_{DER}) + \sum_{n_{BESS}} (P_{BESS}) - P_L = 0, \quad \forall h \qquad (7)$$

$$P_G \in \left( \underline{P}_G, \overline{P}_G \right); \ P_{DER} \in \left( \underline{P}_{DER}, \overline{P}_{DER} \right) \qquad (8)$$

where $h$ denotes hours; $\rho_G$ and $\rho_L$ denote the grid−purchased/sold energy (market) price and load delivered power, respectively; $P_{Gp}$ and $P_L$ are the amounts of grid−purchased/sold and load delivered power, respectively; $n$, $n_{DER}$, $n_{BESS}$ represent the number of all installed/invested units, number of generation units, and the number of BESSs, respectively; $P_{DER}$, $f_{gen}$, $f_{OM}$ indicate the power generated by DERs, the fuel consumption rate of DERs as functions of generated powers, and the operation and maintenance costs of DERs as functions of generated powers, respectively; and $P_{BESS}$, $f_{BESS}$ denote the power delivered by BESS and the cost function of BESSs (to model the battery degradation and life cycle effect), respectively. Constraint (7) maintains the production–consumption balance, and constraint (8) imposes grid−power exchange and DER power generation capacity limits.

In islanded operation mode, the microgrid power supply is mainly offered by RESs and BESS. The output of the whole microgrid should realize the real−time following of load demand changes. In general, there is a large demand for BESS in this operation mode. The EMS takes a load demand, distributed power generation, and SOC of storage battery as input. At the same time, the data are managed and the signals are shown to control the BESS operation. Its control mode mainly exists in the three following situations.

(1) If the difference between the distributed power generation and load demand is positive and the battery is not fully charged, the control system will invoke the charging operation. The remaining power will be distributed among all batteries until the battery is charged to the SOC upper limit. When the battery is charged to the upper limit of SOC, if the microgrid is in the grid−connected mode, the remaining electricity will be sold to the distribution network. If the microgrid is in the islanded operation mode, the control system will invoke the generation power limiting operation.

(2) If the distributed power supply and demand on the load side are equal, there is no need for the BESS to operate, and it may work in standby mode to provide ancillary services such as dynamics frequency and voltage support.

(3) If the difference between the distributed power generation and load demand is negative and the battery is not fully discharged to the lower limit of SOC, the control system will invoke the discharge operation. When the battery is fully discharged to the lower limit of SOC, if the microgrid is in the mode which is connected to the grid, and the remaining required power can be supplied by the distribution grid. If the microgrid is in island operation mode, the load shedding operation will be invoked by the control system.

For battery charge state detection using the neural network method, a large number of corresponding outside factors such as current and voltage, as well as battery charge state data can be applied using training datasheets. The neural network is repeatedly trained and trialed by forwarding the propagation of input information and backpropagation of error transmission. When the predicted charge state is within the error range required by the design, the predicted value of the battery's charge state could be obtained by inputting new data.

The neural network algorithm has a strong nonlinear fitting ability, without considering the internal structure of the battery, for external excitation, the relationship between the input and output can be obtained by training a large number of input and output samples, so it can be that the dynamic characteristics of the battery are well fitted to realize the SOC estimation of the battery. Additionally, it has high estimation accuracy. When there are enough battery data samples, high accuracy can be obtained, and the neural network estimates the SOC. It is suitable for a variety of power batteries and can effectively solve the problems in background technology.

### 2.2. The Structure of the Prediction System

The basic requirement of the prediction system in a microgrid is to forecast the output power of the RESs such as WT on the next day. It is required to forecast not only for the output power individually but also to gather the output power of the whole region. The wind power prediction system should have a good interface with the EMS system. Meanwhile, the wind power prediction system needs to operate in the network of the dispatching system. Its network structure and security protection scheme should meet the requirements of system security protection regulations.

The structure and model of the whole system are shown in Figure 2. Here, are some software functions:

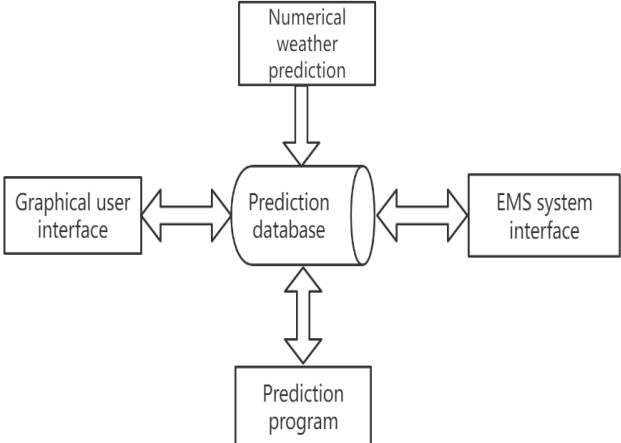

**Figure 2.** Prediction system diagram.

(1) The database of the forecasting system is the core of the system. Each software module closely interreacts through the database. It stores the numerical weather forecasting data from the numerical weather forecasting processing module. Additionally, the predic-

tion program can generate the data of results, the real−time wind power load generated by the EMS system interface program, the wind power data of the market price, etc.

(2) The numerical weather prediction module downloads the weather forecasting data from the system of the weather prediction service provider. After processing, the numerical weather forecast data of each predicted wind farm prediction period is formed and sent to the database of the prediction system.

(3) The prediction module extracts the numerical weather prediction data from the database. The forecasting model calculates the prediction results of the wind power plant and sends them back to the database.

(4) The interface part of the EMS system transmits the real-time power data in every wind turbine to the system database. Meanwhile, the forecasting results are taken from the database and sent to the EMS system.

(5) The graphical user interface module realizes the interactions between the system and users—including complete data and curve display, system management, and maintenance functions.

## 3. BP and GA–BP Neural Network Modeling
### 3.1. BP Neural Network Modeling

The forecasting model of the BP algorithm is shown in Figure 3. The model consists of three parts: the establishment of the BP deep neural network model, training, and prediction. Firstly, modeling serves to analyze the required network topology and the precise number of nodes in each layer of this topic to build a suitable BP network model for this topic. The network is then initialized. Secondly, the relevant network parameters are set and the training function of each layer and the transfer function of each layer are selected. Then, the training of the network begins. If the training result reaches the expected value, it will enter the test part. If the training result does not reach the expected value, it will return to re−train. Then, the network is tested. Finally, the network prediction is made.

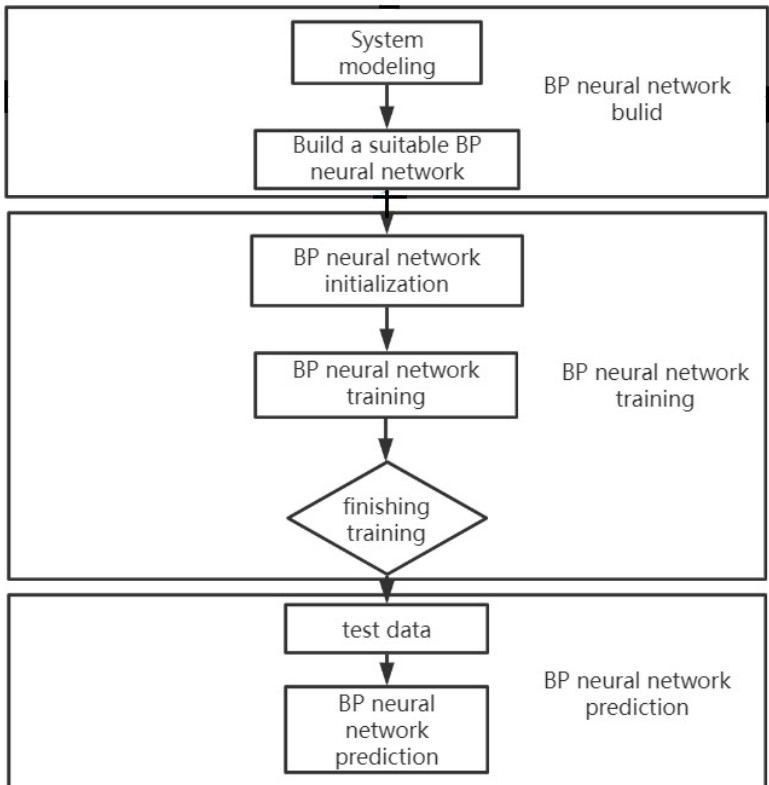

**Figure 3.** BP algorithm prediction model.

BP neural network has a multi−input and multi−output model structure. This can meet any intricate input to output with a non−linear map. By learning the samples, the reasonable solution rules can be automatically extracted, and the complex system can be adaptively modeled. After effective BP network training, the iteration time of the system is less. It can be used in real−time processing. The constructed system also has good robustness and generalization ability. However, a large number of research results show that there are still many unsatisfying aspects of the BP network in both theoretical analysis and practical operation. For example, the traditional BP algorithm uses the gradient descent method. This is a local search optimization method [29]. This method can simply fall into the local extremum during training, leading to the failure of the training. In addition, the BP algorithm is difficult to solve the contradiction between the instance scale and network scale of the application problem. This involves the relationship between the possibility and feasibility of network capacity, that is, the problem of learning complexity. Moreover, the newly added samples should affect the network that has been successfully learned.

The BP network also has the characteristics of "overfitting". In general, the predictive ability of the network is proportional to the training ability. With the improvement of training ability, the prediction ability of the BP network will reach a limit. Additionally, the prediction ability will then tend to decline, which is called "overfitting". Even if there are more samples for network learning, it cannot reflect the internal law contained in the sample. This shows the gap ability to forecasting and the ability to train the network.

To mitigate the limitation of the traditional BP neural network in practical forecasting applications due to the above defects, most people use an optimized algorithm based on the heuristic standard gradient descent method when training the network, including the additional momentum method and adaptive learning rate method—or improved algorithms based on standard numerical optimization, such as genetic algorithm (GA), conjugate gradient method, Bayesian regularization method and Levenberg–Marquardt method (L–M method). The speed of the improved algorithms above varies according to the model complexity, sample scale, network model, and error requirements. In general, the training efficiency of the GA numerical optimization algorithm is higher than that of the heuristic standard gradient descent method.

*3.2. GA–BP Neural Network*

The genetic algorithm (GA) is a heuristic search optimization method based on Darwin's theory of biological evolution, which is used in various applications [30–33]. Selection, crossover, and mutation are its core operations. Three operations are used to screen individuals from the initial population. During the period, the high fitness individuals are left, whereas the low fitness individuals are deleted. Hence, the new group has the relevant information from the previous generation. Additionally, the new group is better compared to the previous generation. The operation is repeated until the conditions are met [32].

The flowchart of GA is introduced in Figure 4, as follows.

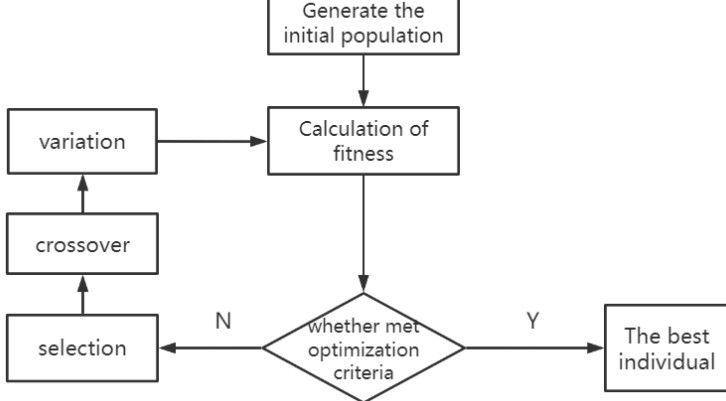

**Figure 4.** Flow chart of genetic algorithm.

The working process of GA is introduced below [33]:

(1) Initialization: calculate the counter of evolutionary algebra as $t$, let $t = 0$; the maximum iteration number is $T$; the initial species population $P(0)$ is $M$ randomly selected individuals. The following problems should be noted when setting the data of GA. For the population size, when the initial population size is set relatively small, the performance will deteriorate and it is difficult to obtain the best result. If the population size is set too high, the computing time of the computer will be prolonged, and the convergence speed will be slowed down, resulting in lower efficiency. Crossover probability plays an important role in genetic algorithms. It can control the frequency of the renewal of individuals in a population. If the crossover probability is too large, the renewal frequency of individuals in the population will be fast, resulting in individuals with high fitness being quickly screened out. If the crossover probability is too small, the search speed will be slow, so the crossover probability is usually set between 0 and 1. For mutation probability, if it is set too high, the search will become random and the convergence will be too slow. If it is too small, the convergence is too fast and it is difficult to produce new excellent individuals, so it is usually set between 0 and 1.

(2) Individual evaluation: calculate the fitness value. The fitness function can calculate the fitness value, which is related to the selected probability individually. The better the fitness value, the greater the chance of being selected and the stronger the viability.

(3) Selection operation: the selection operation can select individuals with good fitness values and pass them on to the next generation. That is, some individuals from the old group are selected for the new group with a specific probability, which is related to the fitness function.

(4) Crossover operation: Crossover operations are core operations. Because of its global search ability, the crossover operation can greatly enhance the searchability of the genetic algorithm. Under the condition of maximizing the structure of chromosomes, two randomly selected chromosomes were partially replaced and recombined to generate new genes, so as to expand the search space. The crossover operator is used to swap and recombine the partial genes of two individuals randomly selected from $P(t)$, to produce new excellent individuals. The crossover operation is shown in the following Figure 5.

cross

A:1100 | 0101 1111 ⟶ A:1100 | 0101 0000
B:1111 | 0101 0000        B:1111 | 0101 1111

**Figure 5.** Crossover operation.

(5) Mutation operation: apply the mutation operator to $P(t)$. An individual is randomly selected from $P(t)$, and some structures of this individual are mutated to obtain a better individual. $P(t)$ is going to obtain a new population $P(t+1)$ after all this. For mutation probability, if it is set excessively high, the search will become random and the convergence will be excessively slow. If it is too small, the convergence is too fast and it is difficult to produce new excellent individuals, so it is usually set between 0 and 1. The mutation operation is shown in Figure 6 below.

mutation

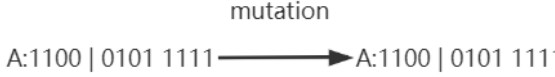

A:1100 | 0101 1111 ⟶ A:1100 | 0101 1111

**Figure 6.** Mutation operation.

(6) Determine the termination condition: if t = t or the result meets the desired standard, then the individual is output as the best solution and the calculation is terminated.

The structure of the BP network is decided by the input–output factor numbers in the sample data. Thus, the length of the individual in GA can be obtained. The network optimization part refers to the optimization and correction of the initial weights and thresholds of the BP network using GA. The defect of the BP network is largely due to its initial weight and threshold being random. Additionally, genetic algorithms can solve this

problem. In the training part of the sample data, the initial weights and thresholds of the BP network are assigned after an individual optimal solution is obtained and following the network's training, testing, and prediction.

## 4. BP and GA–BP Neural Network Realization

The structure of the BP network is determined by the number of input and output influencing factors in the sample data so that the length of individuals in GA can be obtained. The network optimization part refers to the optimization and correction of the initial weights and thresholds of the BP network using GA. Because a large part of the defect of the BP network is due to its initial weight and the threshold value is random, the genetic algorithm can solve this problem. In the training part of sample data, the initial weights and thresholds of BP network are assigned after obtaining individual optimal solutions, and then network training, testing and prediction are carried out.

### 4.1. BP Neural Network Model Design

(1) The input and output nodes are decided upon: this model selects the wind power data of a wind power plant in Seville, Spain, within a total of 96 h in 4 days. Firstly, it is necessary to make the data normalized. The normalization of sample data mainly serves to eliminate the difference in the order of magnitude between the influencing factors, unify the data processing, and improve the prediction accuracy. This topic adopts the normalization method as shown in the following formula:

$$x_{gij} = \frac{x_{ij} - x_{imin}}{x_{imax} - x_{imin}} \tag{9}$$

where $x_{gij}$ is the normalized value of the $j$ sample data of the $i$ influencing factor; $x_{ij}$ represents the $j$ sample data of the $i$ influencing factor; $x_{imin}$ is the minimum value of the $i$ influencing factor; and $x_{imax}$ is the maximum value of the $i$ influencing factor.

For the normalization of the wind degree, the wind direction indicates the direction of the wind, the circle is divided into 360 degrees, and north is 0 degrees. To distinguish wind directions in every degree, the value sine and cosine of wind direction are taken as input. This is to mitigate the large gap between the data affecting the training effect. Selecting the sine and cosine values of the wind direction for normalization is a good choice.

In this model, the average relative temperature, pressure, humidity, wind speed, and sine and cosine wind direction angles are selected as input variables, namely $i = 6$. The real−time electricity price of the wind power plant or wind power load is taken as the prediction target, so the output node number is $k = 1$.

(2) Determine the hidden layer numbers.

The determination of the number of nodes in hidden layers generally requires testing by setting a different number of nodes, comparing the global error of their prediction results, and constantly adjusting the number of hidden layer nodes to test the algorithm. In the actual training, the following formula is generally adopted for a candidate number of hidden layer nodes, and then the number of nodes with the minimum global error is selected as the number of hidden layer nodes.

$$l = \sqrt{m + n} + a \tag{10}$$

where $a$ is the empirical constant, usually $1 < a < 10$, and $m$ and $n$ represent the input number and output number. Through the continuous adjustment of network training, the prediction error is minimized when the number nodes in the hidden layer is 6.

(3) Determine the value of the learning rate.

As an important parameter, the learning rate determines the changes in weights and thresholds in the backpropagation process of the BP neural network. If the learning rate is excessively large, the stable operation of the system may be affected. However, when the model has a small learning rate, the time for training may be directly affected. Hence, the

program will take a long time to converge. Therefore, a learning rate of 0.001 is selected in this model.

(4) Network performance is evaluated by error parameters.

Coefficient of determination ($R^2$):

$$R^2 = 1 - \frac{\sum(\overline{y_i} - y_i)^2}{\sum(\overline{y_i} - \overline{y})^2} \tag{11}$$

Mean square error (MSE):

$$MSE = \frac{\sum_{i=1}^{n}(y_i - \overline{y_i})^2}{n} \tag{12}$$

Root mean square error (RMSE):

$$RMSE = \sqrt{\frac{\sum_{i=1}^{n}(y_i - \overline{y_i})^2}{n}} \tag{13}$$

Mean absolute percentage error (MAPE):

$$MAPE = \frac{1}{n}\sum_{i=1}^{n}\left|\frac{y_i - \overline{y_i}}{\overline{y_i}}\right|100\% \tag{14}$$

Residual prediction deviation (RPD):

$$RPD = \frac{standard\ deviation}{RMSE} \tag{15}$$

where $y_i$ is the actual load value; $\overline{y_i}$ is the predicted load value; and $n$ is the number of historical load data.

*4.2. BP Neural Network Model Analysis*

Figure 7 shows the dataset of six input parameters including the temperature, pressure, humidity, wind speed, sine value of wind angle, and cosine value of wind angle, respectively. It is worth noting that the data of columns 5 and 6 were processed by data normalization. Additionally, there are 96 lines of data which represent 4 days (96 h), where each line stands for 1 h parameters of the wind turbine. For 4 days, the first day and the third day are sunny weather, while the second day and the fourth day are rainy and cloudy. In other words, lines 1–24 and lines 49–72 represent the sunny day, while lines 25–48 and lines 73–96 represent the rainy and cloudy day. This is to distinguish the influence of different weather conditions on the accuracy of load forecasting. In this paper, the six above independent variables of a microgrid in four consecutive days are used as sample data for training, and a load of its microgrid is predicted.

Table 1 presents the data on the power load of the wind turbine. In this model, it represents the output of the model. In the MATLAB simulation module, users can use the relevant functions of network design, training, and simulation provided by Neural Network Toolbox (NNbox) according to their requirements, normalize training samples, develop network initialization, learning rules, and network parameters. In this way, the neural network algorithm learning, and iterative procedures can be realized.

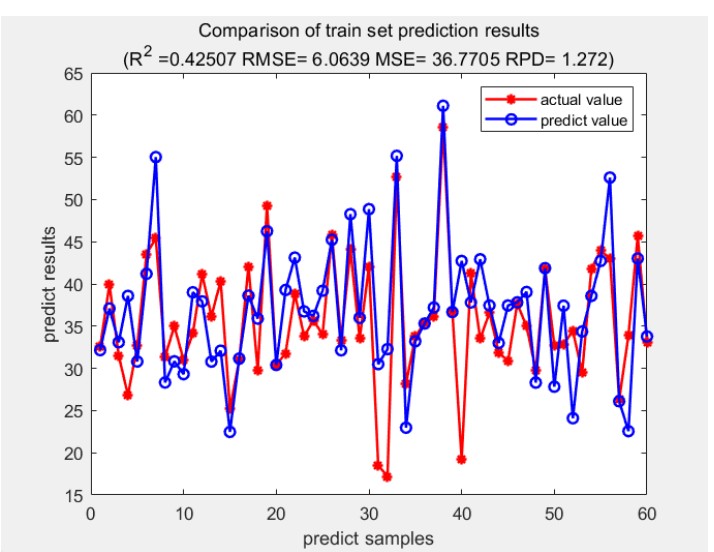

**Figure 7.** Actual value and predicted value in the training set.

**Table 1.** Dataset of the input output parameters.

| | | | | Input | | | |
|---|---|---|---|---|---|---|---|
| Day | Temperature | Pressure | Humidity | Wind Speed | Sine Value of Wind Angle | Cosine Value of Wind Angle | Output |
| 1.00 | 290.16 | 1029 | 55 | 5 | −0.5 | −0.866025 | 1359 |
| 2.00 | 290.66 | 1029 | 55 | 5 | −0.766044443 | −0.642788 | 1404 |
| 3.00 | 290.67 | 1029 | 59 | 4 | −0.766044443 | −0.642788 | 1481 |
| 4.00 | 287.64 | 1029 | 62 | 4 | −0.866025404 | −0.5 | 1577 |
| 5.00 | 285.63 | 1029 | 67 | 5 | −0.866025404 | −0.5 | 1513 |
| 6.00 | 283.63 | 1030 | 71 | 3 | −0.866025404 | −0.5 | 1453 |
| 7.00 | 283.13 | 1030 | 71 | 4 | −0.866025404 | −0.5 | 1577 |
| 8.00 | 283.13 | 1030 | 71 | 4 | −0.939692621 | −0.34202 | 1897 |
| 9.00 | 282.14 | 1030 | 81 | 2 | −0.984807753 | −0.173648 | 2133 |
| 10.00 | 283.14 | 1030 | 76 | 2 | −0.984807753 | −0.173648 | 2081 |
| 11.00 | 282.13 | 1030 | 76 | 5 | −0.866025404 | −0.5 | 2012 |
| 12.00 | 282.14 | 1030 | 81 | 1 | −0.939692621 | 0.3420201 | 1841 |
| 13.00 | 281.64 | 1030 | 81 | 1 | −0.984807753 | 0.1736482 | 1800 |
| 14.00 | 280.64 | 1030 | 87 | 2 | −0.984807753 | −0.173648 | 1882 |
| 15.00 | 280.64 | 1030 | 87 | 4 | −0.939692621 | −0.34202 | 2025 |
| 16.00 | 281.15 | 1030 | 87 | 3 | −0.939692621 | −0.34202 | 2157 |
| 17.00 | 280.14 | 1030 | 81 | 3 | −0.866025404 | −0.5 | 2112 |
| 18.00 | 280.64 | 1030 | 81 | 1 | 0 | 1 | 2021 |
| 19.00 | 280.64 | 1030 | 81 | 3 | −0.984807753 | −0.173648 | 2077 |
| 20.00 | 282.66 | 1031 | 81 | 2 | −0.984807753 | −0.173648 | 2131 |
| 21.00 | 286.16 | 1031 | 71 | 4 | −0.939692621 | −0.34202 | 2319 |
| 22.00 | 288.66 | 1031 | 62 | 4 | −0.866025404 | −0.5 | 2702 |
| 23.00 | 290.66 | 1031 | 55 | 4 | −0.866025404 | −0.5 | 2996 |
| 24.00 | 291.66 | 1030 | 52 | 4 | −0.64278761 | −0.766044 | 3139 |

**Table 1.** *Cont.*

| | | | | Input | | | |
|---|---|---|---|---|---|---|---|
| Day | Temperature | Pressure | Humidity | Wind Speed | Sine Value of Wind Angle | Cosine Value of Wind Angle | Output |
| 25.00 | 292.16 | 1029 | 48 | 3 | −0.64278761 | −0.766044 | 3217 |
| 26.00 | 292.16 | 1029 | 52 | 3 | −0.766044443 | −0.642788 | 3183 |
| 27.00 | 291.66 | 1029 | 55 | 2 | −0.766044443 | −0.642788 | 3092 |
| 28.00 | 288.66 | 1029 | 67 | 2 | −0.866025404 | −0.5 | 3065 |
| 29.00 | 285.64 | 1030 | 76 | 4 | −0.866025404 | −0.5 | 2936 |
| 30.00 | 284.15 | 1030 | 81 | 3 | −0.939692621 | −0.34202 | 2681 |
| 31.00 | 282.64 | 1030 | 87 | 2 | −0.984807753 | −0.173648 | 2592 |
| 32.00 | 282.64 | 1030 | 87 | 4 | −0.939692621 | −0.34202 | 2543 |
| 33.00 | 282.15 | 1030 | 87 | 2 | −0.984807753 | −0.173648 | 2503 |
| 34.00 | 281.64 | 1031 | 87 | 3 | −1 | 0 | 2581 |
| 35.00 | 280.64 | 1031 | 93 | 2 | −0.984807753 | −0.173648 | 2608 |
| 36.00 | 280.15 | 1030 | 93 | 2 | −1 | 0 | 2422 |
| 37.00 | 280.64 | 1030 | 87 | 2 | −1 | 0 | 2217 |
| 38.00 | 279.64 | 1030 | 93 | 2 | −0.939692621 | 0.3420201 | 1875 |
| 39.00 | 279.15 | 1030 | 100 | 1 | −0.984807753 | −0.173648 | 1561 |
| 40.00 | 279.64 | 1030 | 93 | 2 | −1 | 0 | 1213 |
| 41.00 | 279.15 | 1030 | 93 | 2 | −0.984807753 | −0.173648 | 1020 |
| 42.00 | 278.64 | 1030 | 93 | 2 | −1 | 0 | 1103 |
| 43.00 | 279.14 | 1030 | 93 | 3 | −0.984807753 | −0.173648 | 1249 |
| 44.00 | 281.66 | 1031 | 93 | 1 | −1 | 0 | 1323 |
| 45.00 | 285.66 | 1031 | 71 | 2 | −0.984807753 | 0.1736482 | 1286 |
| 46.00 | 288.16 | 1031 | 67 | 2 | −1 | 0 | 1325 |
| 47.00 | 291.16 | 1030 | 59 | 1 | 0 | 1 | 1370 |
| 48.00 | 292.15 | 1029 | 52 | 1 | −0.342020143 | −0.939693 | 1456 |
| 49.00 | 292.66 | 1029 | 52 | 1 | 0 | 1 | 1605 |
| 50.00 | 292.66 | 1028 | 59 | 3 | 0.766044443 | −0.642788 | 1658 |
| 51.00 | 291.66 | 1028 | 55 | 3 | 0.766044443 | −0.642788 | 1592 |
| 52.00 | 288.15 | 1028 | 72 | 1 | 0.766044443 | −0.642788 | 1455 |
| 53.00 | 286.14 | 1029 | 76 | 1 | 0.866025404 | −0.5 | 1489 |
| 54.00 | 284.14 | 1029 | 81 | 1 | 0.325568154 | −0.945519 | 1471 |
| 55.00 | 282.64 | 1029 | 93 | 0 | 0 | 1 | 1418 |
| 56.00 | 281.66 | 1029 | 100 | 1 | −1 | 0 | 1286 |
| 57.00 | 280.64 | 1029 | 100 | 2 | −0.939692621 | 0.3420201 | 1117 |
| 58.00 | 280.64 | 1029 | 93 | 2 | −0.939692621 | 0.3420201 | 973 |
| 59.00 | 280.15 | 1029 | 100 | 2 | −0.866025404 | 0.5 | 891 |
| 60.00 | 280.15 | 1028 | 93 | 1 | −0.984807753 | 0.1736482 | 764 |

**Table 1.** *Cont.*

| | | | | Input | | | |
|---|---|---|---|---|---|---|---|
| **Day** | **Temperature** | **Pressure** | **Humidity** | **Wind Speed** | **Sine Value of Wind Angle** | **Cosine Value of Wind Angle** | **Output** |
| 61.00 | 279.64 | 1028 | 93 | 2 | −0.866025404 | 0.5 | 697 |
| 62.00 | 279.64 | 1028 | 93 | 1 | −0.984807753 | 0.1736482 | 604 |
| 63.00 | 279.15 | 1028 | 93 | 2 | −1 | 0 | 584 |
| 64.00 | 278.64 | 1027 | 93 | 2 | −0.984807753 | 0.1736482 | 617 |
| 65.00 | 278.15 | 1027 | 100 | 2 | −1 | 0 | 673 |
| 66.00 | 278.15 | 1027 | 93 | 2 | −0.939692621 | 0.3420201 | 778 |
| 67.00 | 277.64 | 1027 | 93 | 2 | −0.939692621 | 0.3420201 | 924 |
| 68.00 | 280.15 | 1028 | 87 | 1 | −0.984807753 | 0.1736482 | 1092 |
| 69.00 | 284.15 | 1028 | 71 | 0 | 0 | 1 | 1198 |
| 70.00 | 287.16 | 1028 | 62 | 0 | 0 | 1 | 1240 |
| 71.00 | 289.66 | 1028 | 59 | 1 | 0 | 1 | 1326 |
| 72.00 | 290.15 | 1027 | 55 | 1 | 0.866025404 | 0.5 | 1398 |
| 73.00 | 290.15 | 1027 | 55 | 1 | 0.017452406 | 0.9998477 | 1537 |
| 74.00 | 290.15 | 1026 | 51 | 0 | 0 | 1 | 1792 |
| 75.00 | 291.15 | 1026 | 45 | 0 | 0 | 1 | 2025 |
| 76.00 | 291.15 | 1026 | 45 | 0 | 0 | 1 | 2340 |
| 77.00 | 287.15 | 1026 | 58 | 1 | 0.017452406 | 0 | 2462 |
| 78.00 | 284.15 | 1027 | 71 | 0 | 0 | 1 | 2672 |
| 79.00 | 282.15 | 1027 | 71 | 1 | 0.017452406 | 0.9998477 | 2852 |
| 80.00 | 281.15 | 1027 | 87 | 1 | 0.017452406 | 0.9998477 | 3037 |
| 81.00 | 280.15 | 1028 | 93 | 1 | 0.017452406 | 0.9998477 | 3340 |
| 82.00 | 279.15 | 1028 | 93 | 2 | 0.034899497 | 0.9993908 | 3444 |
| 83.00 | 278.15 | 1028 | 93 | 2 | 0.034899497 | 0.9993908 | 3319 |
| 84.00 | 278.15 | 1028 | 93 | 2 | 0.034899497 | 0.9993908 | 3168 |
| 85.00 | 278.15 | 1028 | 93 | 1 | 0.017452406 | 0.9998477 | 3127 |
| 86.00 | 278.15 | 1027 | 93 | 1 | 0.017452406 | 0.9998477 | 3379 |
| 87.00 | 277.15 | 1028 | 93 | 2 | 0.034899497 | 0.9993908 | 3729 |
| 88.00 | 277.15 | 1028 | 100 | 2 | 0.034899497 | 0.9993908 | 4059 |
| 89.00 | 277.15 | 1027 | 93 | 1 | 0.017452406 | 0.9998477 | 4340 |
| 90.00 | 276.15 | 1028 | 93 | 1 | 0.017452406 | 0.9998477 | 4785 |
| 91.00 | 276.15 | 1028 | 93 | 2 | 0.034899497 | 0.9993908 | 5464 |
| 92.00 | 276.15 | 1029 | 93 | 2 | 0.034899497 | 0.9993908 | 5848 |
| 93.00 | 280.15 | 1029 | 81 | 1 | 0.017452406 | 0.9998477 | 6151 |
| 94.00 | 283.15 | 1029 | 66 | 2 | 0.034899497 | 0.9993908 | 6461 |
| 95.00 | 285.15 | 1030 | 66 | 1 | 0.017452406 | 0.9998477 | 6871 |
| 96.00 | 286.15 | 1030 | 62 | 1 | 0.017452406 | 0.9998477 | 7062 |

The trainable forward inverse feedback function "newff" was introduced in MATLAB to create a BP network. Moreover, the bipolar S−type function "tansig" is the neuron

transfer function in the hidden layer. The pure linear transfer function "purelin" is the output layer neuron transfer function.

After determining the network structure, "newff" will automatically introduce the initialization function "init" to initialize each weight with random default parameters and generate a trainable forward feedback network, which means the return value of this function "net". In MATLAB, network attributes are defined by structures. The function "trainlm" is being used to realize training. The iteration numbers are set to 1000. Additionally, the convergence error is set to 0.01. Part of the code parameters are shown in Algorithm 1:

---

**Algorithm 1**: BP neural network parameters

---

```
Procedure test1()
      temp ← randperm(size(X, 1))
      for i ← 1 to 60 do
            P_train[i] ← X[temp(i)]
      end for
      for i ← 61 to end do
            P_test[i] ← X[temp(i)]
      end for
      M ← size(P_train, 2);
      for i ← 1 to 60 do
            T_train[i] ← Y[temp(i)]
      end for
      for i ← 61 to end do
            T_test[i] ← Y[temp(i)]
      end for
      N ← size(T_test, 2)
      net. trainParam.epochs ← 1000
      net. trainParam.goal ← 1e−3
      net. trainParam.lr ← 0.01
end procedure
```

---

The training set is the data sample that may fit the model. During the process, the training uses the error gradient descent, learning, and trainable weight parameters. The test set can determine the ability to generalize in the final model. This model chooses 1–60 data as the training set and 61–96 data as the test set to check the learning effect. The result of the training and test set are demonstrated in Figures 7 and 8.

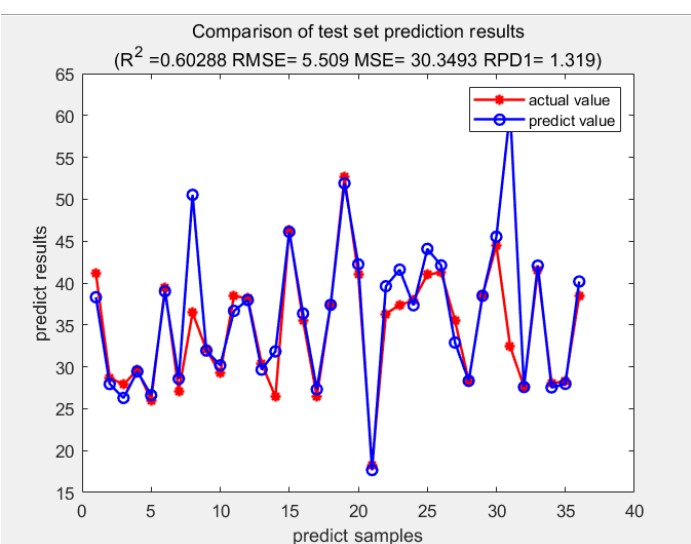

**Figure 8.** Actual value and predicted value in the test set.

In the test set, we can see that the coefficient of determination ($R^2$) = 0.60288; root mean square error (RMSE) = 5.509; residual prediction deviation (RPD): 1.319; mean square error (MSE): 30.3493.

The relatively fast TRAINLM function in a medium−scale network was used. The data show that, although the prediction value and the actual value have a similar trend, the forecasting accuracy is not very good. Therefore, it is essential to find a well−modified algorithm for the problem as well as improve the prediction effect. In addition, the waveform of the prediction curve should become smoother and the prediction error is further reduced.

### 4.3. GA–BP Neural Network Model Analysis

Part of the code parameters are shown in Algorithm 2:

---

**Algorithm 2**: GA–BP neural network parameters

---

```
Procedure test2()
    net. trainParam.epochs ← 1000
    net. trainParam.goal ← 1e−4
    net. trainParam.lr ← 0.01
    net. trainParam.showWindow ← 0
    maxgen ← 100
    sizepop ← 10
    pcross ← 0.8
    pmutation ← 0.1
end procedure
```

---

Crossover operations are the core operations in the GA–BP neural network training. Because of its overall searchability, the crossover operation can greatly enhance the searchability of the genetic algorithm. Under the condition of maximizing the structure of chromosomes, two randomly selected chromosomes were partially replaced and recombined to generate new genes, to expand the search space. In this model, 0.8 is set to the crossover probability and 0.1 is set to the mutation probability.

This model also chooses line 1–60 data as the training set and line 61–96 data as the test set to check the learning effect. The result of the fitness curve, training set, test set, and test set prediction error are demonstrated in Figures 9–12.

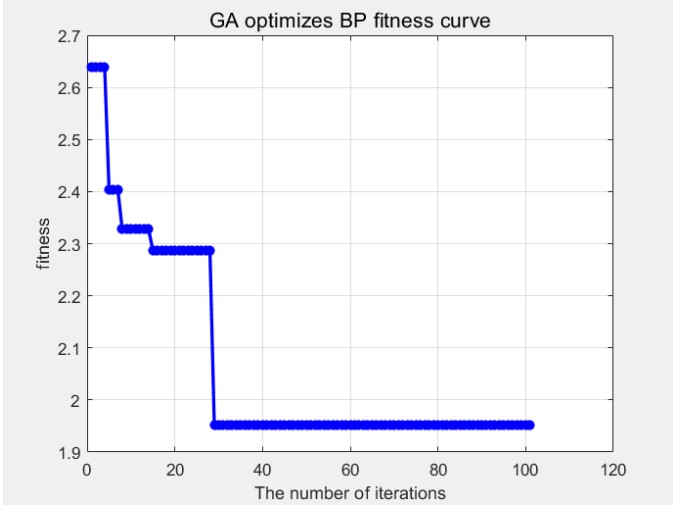

**Figure 9.** GA optimizes BP fitness curve.

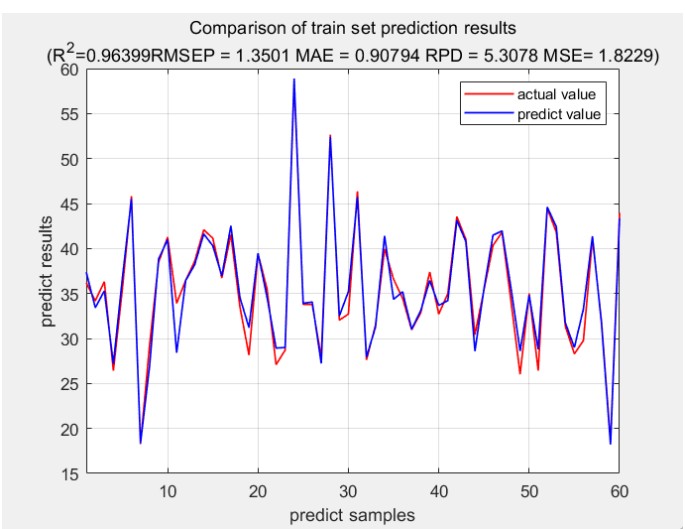

**Figure 10.** Actual value and predict value in train set.

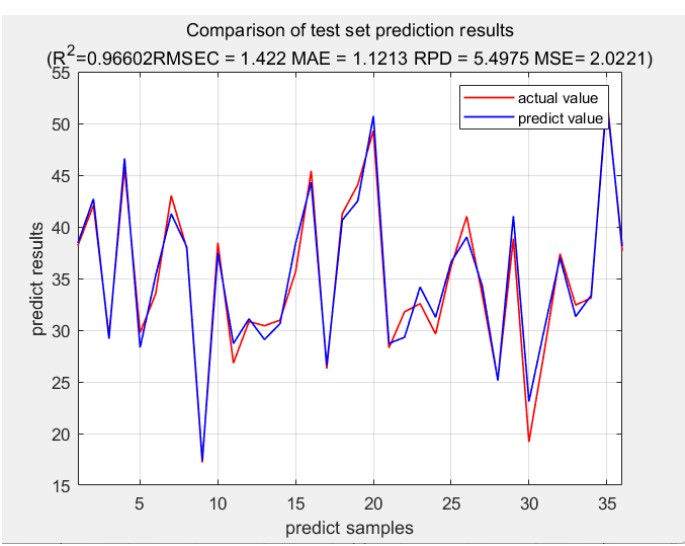

**Figure 11.** Actual value and predicted value in test set.

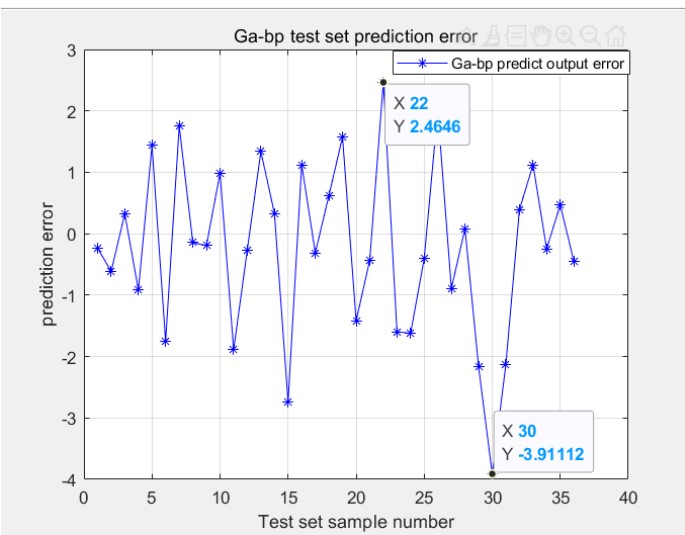

**Figure 12.** Prediction error in test set.

In the test set, we can see that the coefficient of determination ($R^2$) = 0.96602; root mean square error (RMSE) = 1.422; mean square error (MSE): 2.0221; and residual prediction deviation (RPD): 5.4975. Additionally, the test set error is between [−3.91%, 2.4646%]. Such a small error is very accurate.

Table 2 compares the BP and GA–BP neural network errors in the test set.

**Table 2.** BP and GA–BP neural network error comparison.

|  | **MAE** | **MSE** | **RMSE** | **$R^2$** | **MAPE** |
|---|---|---|---|---|---|
| BP neural network | 2.2985 | 30.3493 | 5.509 | 0.6029 | 0.2791 |
| GA–BP neural network | 1.1213 | 2.20221 | 1.422 | 0.966 | 0.0683 |
| Error reduction (%) | 51.20% | 93.30% | 74.18% | 36.30% | 75.50% |

It can be seen from Table 2 that in terms of overall prediction accuracy, the mean square error (MSE) of the GA–BP neural network prediction model is 2.0221, which is significantly smaller than the 30.3493 of the BP neural network prediction model. The error reduction is 93.3%. This shows that the prediction accuracy of the GA–BP neural network prediction model is significantly better than that of the BP neural network prediction model. In terms of the volatility and overall effect of the prediction results, the root mean square error (RMSE) and mean absolute error (MAE) of the GA–BP neural network prediction model are 1.422 and 1.1213, respectively, which are smaller than 5.509 and 2.2985 of the BP neural network prediction model. Additionally, the value of the error reduction is 74.18% and 51.2%, respectively. This shows that the prediction effect of the GA–BP neural network prediction model is also significantly better than the BP and GA–BP neural network prediction model. As for $R^2$, $R^2$ itself is a widely used index to determine the quality of regression models. The value of $R^2$ is larger (close to 1), the better performance of the regression equation fitness. Therefore, it can be concluded that the GA–BP neural network prediction model is better than the BP neural network prediction model in the precision and effect of microgrid load prediction. Moreover, the model with the GA optimized weight threshold has higher prediction accuracy and a faster convergence speed. Additionally, at the point where the BP error is larger, the error decreases more obviously. The results of the GA–BP algorithm are better than those of traditional BP neural networks. The main reason is that GA modifies the random initial weights and thresholds of the BP network. This may make a remedy for the defects of the BP network to some extent. This improves the accuracy of the prediction.

Similarly, wind power and market price can also be used as the output of the model to predict as long as the data corresponding to the input can be obtained. In the machine learning algorithm, the wind power and market electricity price and load are all random data. After training and testing a specific algorithm, the corresponding data will also be obtained. Hence, wind power and market price can also be forecasted by BP and GA–BP neural networks.

## 5. Conclusions

This paper used the reinforcement learning method to study how ANN plays a role in the prediction module of energy management systems in microgrids. Especially using the BP neural network and GA optimized the BP neural network to realize not only load prediction but power and market price as well. After comparing the prediction error in the test set, the results of the GA–BP algorithm are better than those of traditional BP neural networks. The main reason is that GA modifies the random initial weights and thresholds of the BP network. This may make a remedy for the defects of the BP network to some extent. This improves the accuracy of the prediction.

During the debugging part of the BP neural network, data preprocessing is a prerequisite for load forecasting. This will directly affect the accuracy of the prediction. In this paper, first of all, mutation data identification, data correction processing, and data normalization processing are carried out for the collected data to do a good job of preprocessing for later

load forecasting. Then, the error characteristic index of this paper is determined by studying the characteristics of the microgrid. Finally, according to the analysis of microgrid load characteristics in the time dimension, the general variation rule of the load is obtained. This paper established power generation prediction models based on the BP and GA–BP neural networks. The screened historical power generation data were used as samples to make the network training. After comparing the forecasting value and the actual value of the neural network test set and analyzing the error, it was concluded that the prediction performance and precision of the GA–BP neural network are better than those of the BP network.

**Author Contributions:** Conceptualization, C.Z. and M.E.; methodology, C.Z., M.E. and M.L.; software, C.Z. and M.L.; validation, C.Z. and M.E.; formal analysis, C.Z. and M.E.; investigation, C.Z. and M.E.; resources, Z.S.; data curation, C.Z. and Z.S; writing—original draft preparation, C.Z. and M.L.; writing—review and editing, M.E.; visualization, C.Z.; supervision, M.E.; project administration, M.E.; funding acquisition, M.E. All authors have read and agreed to the published version of the manuscript.

**Funding:** This work was supported in part by the Australian Research Council under Grant DP190102501.

**Institutional Review Board Statement:** Not applicable.

**Informed Consent Statement:** Not applicable.

**Data Availability Statement:** Not applicable.

**Conflicts of Interest:** The authors declare no conflict of interest.

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
