# Peer review of "GA−Reinforced Deep Neural Network for Net Electric Load Forecasting in Microgrids with Renewable Energy Resources for Scheduling Battery Energy Storage Systems"

_algorithms, doi:10.3390/a15100338_

Round 1

Reviewer 1 Report

This paper presents a method for predicting the net electric load of a microgrid using a deep neural network so as to realize a reliable power supply as well as reduce the cost of power generation. Genetic algorithm is used to optimize the weights and thresholds of the neural network.

What is the basis of Eq(7)? Some details should be given.

If there are 6 hidden layers, then how many neurons in each of the hidden layer and how are they determined? Some details should be given.

Figure 17 should be a table. Are the values on training data or testing data? Some details should be given.

Author Response

Dear review committee

Thank you for requesting for a revised version of the manuscript. Please enclosed find a point-by-point response to the reviewers’ comments. We are very grateful to the reviewers for their positive and helpful suggestions, and we feel that the quality of the manuscript has been significantly improved as a result. As you will see, we have done our best to incorporate these suggestions as thoroughly as possible.

Sincerely yours,

The authors

Reviewer 2 Report

The authors present the article entitled “GA-Reinforced Deep Neural Network for Net Electric Load Forecasting in Microgrids with Renewable Energy Resources for Scheduling Battery Energy Storage Systems.”

This paper predicts the net electric load of the microgrid using a deep neural network, to realize a reliable power supply and reduce the cost of power generation.

The article presents the following concerns:

  • Abstract section must present quantitative vaules in order to highlight the findings.

  • Line 33: Avoid using apostrophes.

  • Figures must be vectorized to see the details. Figure 4 shows the selection of a frame.

  • Subsection 1.2 must present up to date works. I suggest analyzing the last 4 years. Then, the contribution and the objective must be updated according to the novelty of the work.

  • Please describe the training process in detail. How was the training data selected?

  • Figure 9 and 12 must be presented as Algorithm. Do not provide screenshots.

  • Include a table which compares the findings of the work vs the already report in the literature. 

  • References: There are two [13] references.

  • Figure 17 must be presented as Table.

  • It is recommended to make a brief description of the structure of the text at the end of the introduction.

  • Add hyperlinks to tables, figures, and references.

  • Recommend making a little introduction between points 4 and 4.1

  • Please add  symbols of registered trademarks

  • Line 38-39 can be justified with the following works regarding microgrids in different scenaios: Photovoltaic failure detection based on string-inverter voltage and current signals; Transformerless multilevel voltage-source inverter topology comparative study for PV systems; Leakage current reduction in single-phase grid-connected inverters - A review; A novel integrated topology to interface electric vehicles and renewable energies with the grid; A new predictive control strategy for multilevel current-source inverter grid-connected; Transformerless common-mode current-source inverter grid-connected for pv applications.

  • Line 264-269 can be justified with the following works regarding genetic algoritms in different scenaios: Fuzzy logic and genetic-based algorithm for a servo control system; Non-linear regression models with vibration amplitude optimization algorithms in a microturbine; Self-tuning neural network pid with dynamic response control; A new methodology for a retrofitted self-tuned controller with open-source fpga.

  • Check the Instruction for authors to present the correct format of references.

My biggest concern is that the framework of the state of the art presented in the manuscript is not up to date. This makes the contribution and originality of the manuscript may not be clear. There are several works about the load forecasting model power systems reported in the literature. 

The following misspelling should be checked:

  1. lines 46- 47: your sentence may be unclear or hard to follow. Consider rephrasing by “Its versatility, fast response speed, high energy density, and high efficiency are the main reasons.”

  2. line 250: The phrase “have a tendency” may be wordy. Consider changing the wording by “tend”.

  3. line 394: The numerals “1st”, “3rd” are used instead of the word spelled out. Consider spelling out the numbers.

  4. line 412: “And” may not be the best choice here. Consider replacing it with another word: “Moreover”, “Furthermore”, or removing it.

Author Response

Dear Review Committee

Thank you for requesting a revised version of the manuscript. Please enclosed find a point-by-point response to the reviewers’ comments. We are very grateful to the reviewers for their positive and helpful suggestions, and we feel that the quality of the manuscript has been significantly improved as a result. As you will see, we have done our best to incorporate these suggestions as thoroughly as possible. 

Sincerely yours,

The authors

Round 2

Reviewer 1 Report

The authors have addressed most of my previous comments. From their reply, I realised that they use a single hidden layer neural network with 6 hidden neurons in the hidden layer. There is just one hidden layer, not 6 hidden layers. The authors need to correct this.

Author Response

We would like to thank you for the positive and constructive comments in the first round that helped us to improve the paper in the revision. 

Thank you for your thoughtful comment. We agree with the reviewer. We have fixed this issue in the manuscript.

Reviewer 2 Report

My comments have been addressed

Author Response

Thank you for the positive and constructive comments in the first round that helped us to improve the paper in the revision.